# Exploring Differential Perceptions and Barriers to Advance Care Planning in Dementia among Asian Patient–Caregiver Dyads—A Mixed-Methods Study

**DOI:** 10.3390/ijerph18137150

**Published:** 2021-07-04

**Authors:** Noorhazlina Ali, Philomena Anthony, Wee Shiong Lim, Mei Sian Chong, Edward Wing Hong Poon, Vicki Drury, Mark Chan

**Affiliations:** 1Cognition and Memory Disorders Service, Department of Geriatric Medicine, Institute of Geriatrics and Active Aging, Tan Tock Seng Hospital, Singapore 308433, Singapore; wee_shiong_lim@ttsh.com.sg (W.S.L.); peng_chew_chan@ttsh.com.sg (M.C.); 2Nursing Service, Tan Tock Seng Hospital, Singapore 308433, Singapore; philomena_anthony@ttsh.com.sg; 3Geriatric Education and Research Institute, Ministry of Health, Singapore 169854, Singapore; chongmeisian@gmail.com; 4St Luke’s Eldercare Pte Ltd., Singapore 349326, Singapore; edwardpoon@slec.org.sg; 5School of Nursing, Midwifery and Paramedicine, Curtin University, Perth, WA 6845, Australia; v.drury@curtin.edu.au

**Keywords:** perception of advance care planning, persons with dementia, socio-cultural barriers, mixed-methods study, Asian culture

## Abstract

A parallel mixed-methods study on 20 patient–caregiver dyads in an Asian population was conducted to explore the differential perceptions and barriers to ACP in dementia. We recruited English-speaking patients with mild dementia and their caregivers. A trained ACP facilitator conducted ACP counseling. Patient–caregiver dyads completed pre–post surveys and participated in post-counseling qualitative interviews. We used mixed-methods analysis to corroborate the quantitative and qualitative data. Differential perceptions of ACP were reported among dyads, with caregivers less inclined for further ACP discussions. Post-ACP counseling, caregivers were significantly more likely to acknowledge barriers to ACP discussions than patients (57.9% versus 10.5%, *p* = 0.005). Thematic analysis of the interview transcripts revealed four themes around barriers to ACP: patient-related factors (transference of decision making, poor cognition and lack of understanding, and dis-inclination to plan for the future), caregiver-related factors (perceived negative impact on the patient, caregiver discomfort, and confidence in congruent decision making), socio-cultural factors (taboos, superstitions, and religious beliefs), and the inappropriate timing of discussions. In a collectivist Asian culture, socio-cultural factors pose important barriers, and a family-centric approach to initiation of ACP may be the first step towards engagement in the ACP process. For ACP in dementia to be effective for patients and caregivers, these discussions should be culturally tailored and address patient, caregiver, socio-cultural, and timing barriers.

## 1. Introduction

Advance care planning (ACP) is defined as a process that supports individuals in understanding, exploring, discussing, and documenting their personal values, life goals, and preferences for future medical care, and entails communication with loved ones, caregivers, or healthcare professionals [1]. ACP was included in the World Health Organization’s Global Dementia Observatory Framework [2], which strives to increase awareness in dementia as a public health priority and to advocate for action at national and international levels. One of the global action areas under this framework was to improve end-of-life care in persons with dementia (PWD) by promoting awareness on ACP, respecting the values and preferences of PWD, and empowering PWD to make choices about their care. ACP encourages people to engage in discussions about future health choices and medical care. From the patients’ perspective, ACP prepares them for death and dying, allows them to exercise control, and relieves loved ones of the burden of decision making [3,4,5].

There has been an upsurge in research publications in the last decade on ACP for persons with dementia [6]. However, compared to individuals with a chronic medical illness, such as heart failure and end-stage renal disease [7,8], the effectiveness of ACP in PWD on clinical outcomes has not been extensively studied [9,10], with limited examination and synthesis of scientific evidence for improving ACP in dementia [11]. This group of vulnerable individuals is at risk of decisional incapacity given the natural course and progression of the dementing illness. Due to the length of illness, caregivers of PWD also often experience prolonged burden in their caregiving role that places extreme stress on prior relationships [12]. Thus, ACP when conducted in the early stages of dementia can be an important gift of legacy for the family. Although the optimal timing for ACP in dementia remains to be established, the earlier (mild and moderate) stages afford a limited window of opportunity when PWD are still able to indicate their preferences and designate their preferred surrogate decision maker [13].

Despite the many benefits of administering ACP early in PWD [14,15,16], there are barriers encountered prior to engaging them for ACP discussion, viz, uncertainty in decision making, existential and emotional barriers, timing of ACP discussions, lack of understanding of ACP, distrust in healthcare professionals, and unequipped healthcare systems [17,18,19,20]. A conceptual framework has been proposed that aligns these perceived barriers to the different steps of the ACP process, namely, pre-contemplation, contemplation, preparation and values clarification, actions and maintenance, or reflection on one’s choices [21]. Using this conceptual framework, the care recipient-, caregiver-, and physician-related barriers to contemplation, discussion, and documentation have been reported at various steps of the ACP process [22,23,24,25,26,27]. The current model addresses the general and common barriers to ACP with particular emphasis on care recipient-related barriers to promote health behavior change [22].

Emerging literature highlights the unique influence of socio-cultural factors above and beyond conventional barriers in the contemplation stage, thus hindering the actual initiation of ACP in dementia [25,28]. For instance, in the Chinese culture, it is often considered taboo to discuss matters pertaining to death as this can be deemed inauspicious or even disrespectful [29,30]. Dementia itself is also viewed as a taboo topic, hence Asians are reticent to discuss it. Another important barrier to ACP within the Asian context is that Asians generally adopt a rather passive attitude in the management of their illnesses, often opting to leave decision making on end-of-life treatment to their family members [31,32,33,34]. This contrasts with the Western culture, which places emphasis on individual autonomy in decision making [35].

With dementia and ACP both highly regarded as taboos in the Asian culture, the conduct of ACP discussion has to be disease-specific and culturally tailored [36], before any intent or contemplation can occur to promote a behavioral change towards initiating ACP. In addition, the impact of socio-cultural influences on care recipient or caregiver perceptions of ACP in dementia in Asian populations has hitherto not been systemically studied and the influence of spirituality, religion, and traditional Chinese philosophy on ACP is not well understood [37].

ACP in Singapore was first introduced by Respecting Choices of Wisconsin, United States of America, in 2009 [38]. Singapore, a Southeast Asian country, is a multicultural, multi-ethnic, and multireligious society with English being the country’s default cultural lingua franca that unite various ethnic groups. Chinese Singaporeans make up the majority of the population (76.2%), followed by Malays (15%), ethnic Indians (7.4%), and Eurasians. Singapore’s healthcare system is traditionally paternalistic [30], such that the authority of physicians is often not disputed, and healthcare decisions are medically guided. The population’s cultural and ethnic diversity can explain the influence of traditions and beliefs on healthcare utilization behavior and medical decision making. ACP was a novelty then and was initiated in people with advanced illness, such as advanced heart failure and respiratory disease. In 2011, a national ACP program (which was later renamed as Living Matters) was established and there was a progressive nationwide adoption of ACP over the last decade. However, there is still no structured dementia-specific or dementia-focused ACP to date.

We conducted a pilot project of ACP discussions with patient–caregiver dyads early in the course of dementia in a memory clinic in Singapore. We aimed to (a) examine the disparity in perceptions among subjects with mild dementia and their caregivers towards ACP after undergoing ACP counseling; and (b) explore the barriers to ACP in dementia.

## 2. Materials and Methods

### 2.1. Study Design

A parallel mixed-method study design was conducted, whereby the quantitative and qualitative data were collected concurrently. Mixed-methods research is underpinned in the pragmatic research paradigm; its comprehensive approach and employment of diverse data sources yield a broader picture and more comprehensive support for validity in the investigation of an underlying phenomenon, such as ACP, in “real-world” practice [39]. In our study, we wanted to obtain different yet complementary data to examine the perceptions of PWD and their respective caregivers on ACP, which was then a novelty in Singapore. The central phenomenon to be explored was the perceived barriers to the initiation of ACP discussions in PWD and their caregivers in a memory clinic setting in Singapore.

### 2.2. Study Population

Subjects were recruited from the outpatient attendees of the Memory Clinic at a tertiary hospital over a period of 2.5 years (September 2010 to February 2013). We recruited patients who were newly diagnosed with dementia based on the *Diagnostic Statistical Manual, 4th Edition* criteria and also fulfilled the following conditions: mild severity, as defined by a global Clinical Dementia Rating score (CDR) of 0.5–1; no significant mood disorders or behavioral problems; and had an identified caregiver who was conversant in English (as all sessions were conducted predominantly in the English language). We excluded those with moderate or severe dementia (CDR ≥ 2) or who had a concomitant diagnosis of delirium, as they might not be able to fully understand or participate actively in the conversations pertaining to ACP due to the novelty of ACP in Singapore then. Those with unstable medical conditions, defined as more than 2 hospitalizations in the last 1 month prior to recruitment, were also excluded since they were still in recovery from acute episodes of their illness.

Participant information sheets were given to eligible patient–caregiver dyads, as assessed by their primary physicians, prior to consent taking, and participation in the study was entirely voluntary. Informed consent was obtained from patient–caregiver dyads in the comfort of a private room within the hospital. Both patients and caregivers were encouraged to ask questions and clarify any doubts that they might have prior to consenting to the study. All study participants were assigned unique code numbers to maintain anonymity and confidentiality. All survey data and interviews were de-identified and stored securely with restricted access by the study group. The conduct of the study was approved by the institutional review board of the National Healthcare Group (NHG DSRB Ref: 2010/00396).

### 2.3. Data Collection

We collected patients’ demographic data, including age, race, gender, education level (years of formal education), comorbidity burden (measured using Charlson’s Comorbidity Index), functional data of basic activities of daily living (Barthel Index), instrumental activities of daily living (Lawton and Brody’s IADL), cognitive assessment (modified Chinese Mini-Mental State Examination), and measures of caregiver burden (22-item Zarit Burden Interview). We also collected caregivers’ demographic data, such as age, gender, years of education, relationship, and co-residence with patient.

### 2.4. ACP Counseling

The dementia nurse (P.A.), a trained ACP facilitator, scheduled an ACP counseling session with the patients and their respective caregivers after they had been apprised of the dementia diagnosis. The content of the session was tailored towards ACP discussion in the context of dementia and entailed educating patient–caregiver dyads on ACP, exploring their fears and concerns in initiating ACP, and discussions on preferred future care plans. The session was conducted in the form of a semi-structured interview and usually lasted for an hour. A short debriefing session was arranged to address any concerns or queries that might have surfaced during the counseling session and follow-up plans were carried out if required.

### 2.5. Pre–Post ACP Counselling Questionnaires

Each patient–caregiver dyad was given a questionnaire before and after the ACP counseling session and they were surveyed individually. We modeled the framework of our questionnaire after a study by Fried, Bradley, Towle, and Allore [40] and developed a culturally acceptable questionnaire. The pre–post counseling questionnaires assessed the patient–caregiver dyads’ perceptions of ACP, and explored their understanding of the purpose of basic ACP and their attitudes and feelings about end-of-life communication with each other, in particular, their understanding of the illness, willingness to have further discussions, and the acknowledgement of barriers to communication. The post-counseling questionnaire also assessed their experience during ACP counseling.

### 2.6. Qualitative Interview

Either one of the two principal investigators (M.C. or N.A.) met up with the patient–caregiver dyad a minimum of a week later (allowing up to 4 weeks if a delay in scheduling occurred) after the ACP counseling session. The qualitative interview occurred at the patient’s place of residence or in a private room in the hospital. It usually lasted between 20 min to 1 h. An audiotape recorder was used to record the entire interview and conversations were then transcribed verbatim. The interview aimed to accomplish the following objectives: (a) to enable the subjects and caregivers to elaborate further on their perceptions and feelings with regards to ACP; and (b) to identify barriers to ACP discussion.

### 2.7. Quantitative and Qualitative Data Analysis

We employed concurrent collection, analysis, and interpretation of the quantitative and qualitative data for this study [41]. We conducted descriptive analyses of the baseline patient and caregiver characteristics, and univariate analysis by Fisher’s exact test to determine the differences between the patient–caregiver dyads’ responses to the questionnaires; in particular, for this paper, the end-of-life communication between patients and caregivers. Statistical Package for the Social Science software version 19.0 was used for quantitative analysis and a *p* value of <0.05 was considered to be significant. Results from the quantitative analysis helped to inform the interpretation of the qualitative data about barriers to ACP.

For the qualitative data analysis, data from the interviews were analyzed using thematic analysis by two qualitative researchers (E.P.W.H. and V.D.). Thematic analysis [42] is a process for encoding data that involves breaking down of data into codes and then organizing codes data into thematic sets. It involves “searching across a data set to find repeated patterns of meaning”, to allow themes to emerge from the data via an approach that is interpretive, descriptive, and inductive.

E.P.W.H. and V.D. coded the data from the transcripts separately using line-by-line coding and initial codes were generated. Anonymity of the transcripts was assured with removal of patient–caregiver dyads identifiers. Both researchers met to compare the initial codes and an agreement was reached between them to decide on the final focused codes. Any disagreement was resolved through consensus. The focused codes were then raised to conceptual categories, and themes that were generated were subsequently compared with the literature obtained prior to the interviews. Memos were written to determine the data that needed to be collected during subsequent rounds of data collection to enhance the understanding and insights that were gathered from previous interviews. The selected themes and quotes were reviewed and endorsed by the rest of the research team.

Integration of both quantitative and qualitative data into a single database enabled the research team to further analyze and simultaneously interrogate the data with equal weightage to examine the patient–caregiver dyads’ perceptions of ACP and explore the barriers to initiation of ACP.

## 3. Results

### 3.1. Characteristics of the Patient–Caregiver Dyads

We identified 30 patient–caregiver dyads for the study, of which 20 agreed to participate. Reasons cited for non-participation include ‘not keen’, ‘too busy’, ‘don’t want to talk about ACP’, and ‘worried if the discussion will cause patient to be depressed’. Among the 20 patient–caregiver dyads who underwent ACP counseling, two declined further interviewing (yielding 18 completed transcripts) and one did not complete the post-counseling questionnaire (yielding 19 completed pre–post questionnaires).

Table 1 depicts the baseline characteristics of the 20 patient–caregiver dyads. Patients who underwent ACP counseling had a mean age of 75.9 years, were predominantly male, well-educated, and of Chinese ethnicity. Their mean CMMSE score of 22.8 corresponded to their mild stage of dementia, predominantly of Alzheimer’s disease etiology. They were not functionally impaired, as is evident in the mean score of 14.9 on the IADL scale, and were healthy with few comorbidities based on their low Charlson Comorbidity Index score of 4.6. The majority of caregivers were males, highly educated, and either adult children (55%) or spouses (40%).

### 3.2. Disparity in Patient–Caregiver Dyads’ Perceptions on ACP

Prior to ACP counseling, only 30% of the patients and 50% of the caregivers had heard of ACP. Although 70% had thought of their healthcare proxy, only 30% of the patients had made future care plans prior to ACP discussion.

There were notable differences in perceptions between the PWD and caregivers, as shown in Table 2. Before ACP counseling, 85% of patients surveyed agreed with the importance of communicating to their caregivers about their illnesses and 55% would like further discussion with their caregivers. After ACP counseling, more patients (68.4%) were keen for further discussions despite a reduced proportion who understood the importance of talking about their illness. In contrast, despite undergoing ACP counseling, a higher percentage of caregivers (57.9% post-counseling vs. 45% pre-counseling) acknowledged barriers to communication with a concomitant decline in the number of caregivers (85% to 78.9%) who were keen to explore ACP further. Post-ACP counseling, caregivers were significantly more likely to acknowledge barriers to ACP discussions than patients (57.9% versus 10.5%, respectively; *p* = 0.005, Fisher’s exact test). Compared to patients, caregivers were also more likely to find ACP useful in understanding complications of illness (*p* = 0.042, Fisher’s exact test). Though more caregivers (78.9% versus 68.4% patients) found ACP helpful in exploring their preferences for future treatment, the difference was not statistically significant (*p* = 0.714, Fisher’s exact test).

### 3.3. Barriers to ACP

Thematic analysis of the transcripts revealed several barriers to ACP (Table 3). They can be categorized into patient-related factors, caregiver-related factors, socio-cultural factors, and the inappropriate timing of the discussions.

#### 3.3.1. Patient-Related Factors

(i) Transference of decision making to others

In our Asian culture, filial piety and family togetherness are greatly emphasized. Not surprisingly, elderly patients tend to take on the dependent role and entrust their children with the burden of care and with any decision making about future care plans. This transference of care was evident in some respondents who indicated that they would readily leave family members to liase with the healthcare team about treatment decisions. This reinforces their own dependent role in the decision-making process, and effectively transfers the burden of decision making to the proxy family member. Inevitably, this downplays the perceived need for discussion with caregivers about their personal preferences for future treatment.

“*Well that’s left to the children how best they can look after me and to make sure that I have the best treatment although they know it may be terminal*”(Patient 1)

(ii) Poor cognition and perceived lack of understanding

Poor cognitive functioning and the ensuing perception that patients might not understand or be able to engage in advance care planning was viewed as a reason for not discussing ACP.

“*Yes there is a barrier but the problem is that um she’s not able to actually understand what we are asking*”(Caregiver 9)

“*No I don’t talk to her about the future because you know I can talk to her [about] the future then [the] next day.. I’ll tell her.. did you remember.. she’ll said no*”(Caregiver 10)

(iii) Lack of inclination to plan for the future

When asked about their willingness to hold further discussions with their caregivers about their future care, most of the patients did not see the necessity. They cited difficulties in planning for their future due to the uncertainty in the manner and timing of disease progression. There was no urgency in initiating ACP discussion or plan for the future since there was no compromise in their present health status. Instead of being pre-emptive, they were quite content to adopt a more passive wait-and-see approach. Some of them displayed imperturbable and cavalier attitudes towards ACP: “Because nothing happen to me”; “just be happy”; “there is not much feeling”; “my condition is okay I’ve got nothing to worry about”; “Nothing. It’s better to forget all the unhappiness”; “Those are not important to me”.

#### 3.3.2. Caregiver-Related Factors

(i) Perceived negative impact on patients

Although the majority of caregivers described the session as being important, one caregiver was concerned that discussion of end-of-life issues would cause anxiety in patients and another caregiver felt that such sessions might not be appropriate for everyone. A concern was voiced by some caregivers that discussing such issues would result in further deterioration of the patient’s health and would negatively influence the patient’s decisions.

“*I think they are afraid.. when they don’t want the sick(ness) to get worse they’re afraid and we all too will get afraid I think as I said if I get round to this stage where I have to go on life support…*” (Caregiver 16)

(ii) Caregiver’s discomfort

There was a certain degree of discomfort and difficulty in broaching the topic of ACP to their loved ones among the caregivers in view of the sensitivity of the issues that may arise during the discussion.

“*I’m very glad that this session has taken place and he has been very frank and open and now I also know….his wishes because it’s a very sensitive topic and it is even though we love one another it is very difficult to approach you know*” (Caregiver 1)

(iii) Confidence in congruent decision making

The majority of caregivers agreed that decision making on the care of the patient would be done through consultation with other family members and the doctor. There was a general consensus that, as caregivers, they know the care recipients well, and hence would be able to make decisions that were congruent and in keeping with the latter’s choices and in their best interest, even without any prior exploration or discussion.

#### 3.3.3. Socio-Cultural Factors

Cultural beliefs were commonly identified as one of the main reasons for not discussing further about advance care planning. Overwhelmingly, caregivers reinforced the belief that ACP should not be initiated, as such discussions of end-of-life issues were considered taboo and run counter to prevailing Asian cultural values, such as filial piety and respect for the elderly. Some caregivers cited superstitious beliefs surrounding such discussions; for instance, how such discussions were tantamount to bringing bad luck and cursing their loved ones.

“*Taboo lah okay I think because of the Asian culture.. taboo.. death is taboo to them*” (Caregiver 10)

“*No because we Chinese ah don’t want to mention all this*” (Caregiver 7)

The majority preferred to allow their spirituality and religious belief to determine how they would subsequently journey in life with their illness. Taking an active role in planning for future deterioration in their condition appears to be at odds with the beliefs and worldview of religious piety, which emphasizes submitting to the sovereignty of the Almighty.

“*I believe that my hands my life is in the hands of my God so whatever happen uh he will provide for.. for my safety and welfare*” (Patient 3)

“*The life is not mine. the life I believe is given to me by the Almighty*” (Patient 13)

#### 3.3.4. Inappropriate Timing of Discussion

An indirect corollary of the patient, caregiver, and socio-cultural barriers described above, is an inertia among caregivers against initiating ACP discussions in the mild stage of dementia, despite acknowledging the benefits of conducting these discussions prior to loss of cognitive functioning. Thus, some caregivers alluded to this inappropriate timing of the discussion by rationalizing that the patient’s condition had not yet sufficiently deteriorated to such an extent as to warrant discussion of such “serious” and taboo matters.

“*We feel that our dad’s condition our parents’ condition whatever illness they have or whatever is still within control it hasn’t reached a stage where it has to be uh has to be seriously discussed*” (Caregiver 1)

## 4. Discussion

### 4.1. Differential Perceptions of ACP among PWD and Caregivers

Our study adds to the growing body of evidence on ACP discussions in dementia by reporting the differential perceptions of ACP among patient–caregiver dyads of predominantly Chinese ethnicity in an Asian setting and the somewhat unexpected positive response towards ACP by PWD.

Contrary to most published Western literature of increased receptivity to ACP in both patients and caregivers [43,44], the proportion of caregivers in our study who were keen on further ACP discussions declined although the proportion who agreed to the importance of ACP remained constant after ACP counseling. This contrasted with the patient group, where there was a post-counseling increase in the proportion who was keen to have further ACP discussions despite a reduction in those who deemed this important. The observed post-counseling increase in the proportion who acknowledged barriers to communication in both groups, could be attributable to increased awareness of perceived barriers as a result of ACP counseling. Interestingly, this was associated with a differential response to willingness for further discussion, being increased amongst PWD and decreased among caregivers. The differential perceptions may pose a barrier to continued ACP discussions, given that ACP is a dynamic process that involves engagement of conversations in an iterative manner, reflections, and values clarification.

In this study, persons with dementia reported positive responses towards ACP, where the majority found it useful in understanding the complications of an illness and exploring preferences for future treatment, and were willing to have continued ACP discussions. This finding is in contrast to a study by Dening et al. [45], where PWD did not perceive the potential value of ACP, and to an integrative literature review by Geshell et al. [46], where many PWD held a neutral to negative view towards ACP. Engagement in ACP discussions can be cognitively demanding for PWD and they may have difficulties in understanding how articulation of healthcare preferences may influence care later. Nevertheless, the keenness and readiness for ACP by PWD may represent the pre-contemplation stage of the ACP process and the first step towards the contemplation of treatment wishes and values in the event of decisional incapacity. Meaningful participation in ACP by PWD can be supported and facilitated by utilizing decision-making tools or aids (clinical vignettes, narratives, illustrations, and video clips), to frame the conversations in a structured and intelligible manner [47,48,49].

### 4.2. Barriers to Initiation of ACP

#### 4.2.1. Patient-Related Barriers

Overall, the results of this study affirm the linking of barriers to the steps of contemplation and discussion in the ACP process. This is congruent with the conceptual model of Schickedanz et al. [22], who demonstrated the self-identified barriers to ACP. The reported patient-related barriers in this study are consistent with other research studies [50,51] examining the factors that hinder initiation of ACP in PWD or early cognitive impairment. The transference of decision making to other family members may be viewed as the proper social order to maintain the familism and conduct of family duties, and a showcase of filial piety in Asian culture [37]. This inevitably propagates the proclivity of not planning for the future should the patient’s health status deteriorate to the point of incapacity. The cognitive deficits experienced by PWD may preclude them from fully participating in ACP conversations due to the lack of understanding of what ACP entails and lack of appreciation of the future end-of-life care, as reported in this study. Nevertheless, people with cognitive impairment can still participate meaningfully in ACP conversations, with the listener being vigilant to discern their preferences and values [52].

#### 4.2.2. Caregiver-Related Barriers

We expanded on the conceptual model of the process of ACP [22] by including insights from mixed methods analysis on caregiver-related barriers, which feature prominently in our study. Caregivers were more likely to acknowledge barriers to ACP than patients with mild dementia. Although caregivers were also more likely to find ACP useful in understanding the complications of illness, they revealed their discomfort in broaching end-of-life issues due to the perceived negative impact of such discussions on patients, which is consistent with past studies [53,54]. A unique finding in our study was the perceived congruence in future care decisions by the caregivers. This may help explain the reluctance among caregivers to engage in further ACP discussion post-counseling. Such perceptions may result in avoidance behavior to engage in ACP as caregivers attempt to navigate through existential tensions confronted during discussions on end-of-life care as dementia progresses to the terminal stage in their loved ones [19]. Furthermore, incongruent end-of-life care preferences have been reported between patients, family and physicians [55,56]. A cross-sectional study in the UK [57] reported less agreement for future hypothetical health states between PWD and their family caregivers in the choice of end-of-life treatment. Batteux, Ferguson, and Tunney [58] provided further evidence of surrogate inaccuracy by reporting discrepancies between a surrogate’s choices and those made by cognitively intact care recipients. Further studies are needed to ascertain whether the confidence of caregivers in congruent decision making is borne out in actual practice.

#### 4.2.3. Collectivist Culture Influence Patient–Caregiver Dyads’ Perception of ACP and Socio-Cultural Barriers to ACP

Our results reinforce the central role of socio-cultural factors in influencing ACP discussions. Consistent with the esteemed values of the Asian culture, the acceptance of decision making by the caregiver (usually adult children) on behalf of the infirmed parent is an unspoken responsibility akin to an act of filial piety. There seems to be an implicit understanding that caregivers will know the preferences and wishes of PWD though no prior discussions had occurred. Our findings concur with two local studies that highlight the influence of the family unit in the decision-making process [54,59], thus reinforcing the prominence of a collectivistic culture among Asians [60,61]. A family-centered ACP model, involving PWD and their caregivers, also has been reported to be suitable for Asian countries with a predominantly Chinese culture, where relational autonomy often times supersedes personal autonomy [62].

In a collectivist society with pervasive cultural differences, such as Singapore, culture has a direct influence on communication [63]. Topics such as death, end of life, and sickness are not explicitly or frequently discussed; thus, it is important to maintain sensitivity towards such topics when discussed. Our study highlighted the contribution of cultural diversity as a communication barrier to ACP, such as the twin taboo of dementia and ACP; superstitious beliefs surrounding ACP discussions; and spiritual and religious beliefs. A person’s culture shapes the way they think and how they make meaning out of illness, suffering, and dying [64]. Hence, culture inevitably plays an important role in influencing decision making towards the end of life [65,66]. There have been calls to develop a culturally tailored ACP to meet the needs of culturally diverse populations [37,67].

#### 4.2.4. Timing of ACP Initiation and Discussion

In our study, the patient, caregiver, and socio-cultural barriers combined to create a negative perception, especially among caregivers, regarding the inappropriate timing of initiating ACP discussions at earlier stages of dementia. Patient–caregiver dyads viewed good health and early stage of dementia as indicators to delay ACP discussions. This is tantamount to a retrograde shift from the contemplation/discussion phase to the pre-contemplation phase. Despite scientific publications acknowledging the time-sensitive need for ACP in PWD [46,68], there is currently no consensus on the optimal timing to initiate ACP discussions during the course of dementia [28]. Compared to the situation in cancer patients, where ACP discussions are initiated when there are clear demarcations between curative and palliative treatment, persons with dementia go through a less predictable disease course [14]. As a result, ACP discussions can only be initiated when patients and caregivers are ready to be engaged in such sensitive dialogues, hence emphasizing the need to tailor such discussions to individual readiness and openness. Nevertheless, beginning ACP early in the course of dementia can provide opportunities for ongoing and gradual conversations to slowly engage patients and caregivers in participating in the planning process [69].

A possible complex array of factors need to be considered to determine the optimal timing of ACP discussion in dementia, such as acceptance of the diagnosis of dementia, understanding of the trajectory of illness and future cognitive decline, decision-making capacity, readiness to engage in emotionally laden conversations, and the ability to decide on preferences based on future clinical scenarios. Hence, further research is required to determine the optimal timing for ACP discussions in PWD and concurrently examine the factors that contribute to initiation of ACP along the dementia trajectory.

### 4.3. Current State and Progress of ACP in Singapore

There was an increase in the number of completed ACP documents in Singapore over the last decade, from 3 in 2011 to about 14,000 in 2018 [38]. From a hospital-centric practice perspective, initiation of ACP discussions are typically among individuals with life-limiting illnesses, such as advanced cancer or terminal illness, but ACP discussions are now being moved upstream to the primary care setting for individuals with chronic diseases and who are healthier. There are three tiers of ACP discussion promulgated in Singapore [70], depending on the individual’s medical condition: (1) general ACP for relatively well and healthy adults; (2) disease-specific ACP for patients with progressive, life-limiting illnesses with frequent complications; and (3) a Preferred Plan of Care for patients with advanced illness with a prognosis of less than one year. Disease-specific ACP forms meant for end-stage renal failure, advanced heart failure, and chronic lung disease were developed, but what is notably absent currently is disease-specific ACP forms for dementia. Given Singapore’s collectivist society and multicultural communities, local research in the conduct of ACP discussions among PWD is surprisingly still lacking. Advance care planning has gained wide local recognition as one component of multi-faceted dementia care. It does not encompass solely the completion of advance directives, but more importantly, the initiation of discussions and contemplation of sensitive topics that people tend to avoid. Our study attempts to address the dearth of local research in ACP for PWD and in explicating our research findings, we hope to advocate for progress in initiation and documentation of ACP discussions for PWD through strategies that consider the patient–caregiver dyads’ perception in order to overcome barriers to ACP.

### 4.4. Limitations and Strengths of the Study

There were notable limitations in our study. The study population was not a true representative of the diverse ethnic variation in Singapore, since there were no Malays or Eurasians recruited. Even among recruited subjects, our English-speaking sample may be reflective of a more privileged group who may not represent the general population; nonetheless, the more reserved views to ACP reported in this study are consistent with the prevailing socio-cultural values. An additional challenge for the analyses in the current study was the semantic inaccuracies amongst our local elderly English language speakers. As most Singaporeans express themselves using a combination of English and their mother tongue during casual conversation, much effort had to be made to discern the underlying themes from the recorded conversations. However, we believe this may also contribute to the richness and diversity of ideas in the context of a multi-cultural society.

This study also highlighted the unique influence of socio-cultural factors above and beyond the identified caregiver and patient-related barriers, which together contrived to yield a timing barrier in the context of ACP discussions in mild dementia. An added strength of the study was the complementarities afforded by using a mixed-methods analysis, which allowed for a broader and more comprehensive explication of this phenomenon to cross-validate and complement individual findings.

## 5. Conclusions

Advance care planning provides an opportunity for PWD to articulate their preferences and values while still retaining decision-making capacity but not without confronting barriers to contemplation or initiation of such emotionally sensitive conversations. Our study highlights the central roles that family and culture play in ACP discussions in our Asian society in tandem with patient and caregiver barriers to effective ACP communication. Differential perception towards ACP may hinder PWD and caregivers from taking the first step towards initiating intimate conversations on future care. Future studies are needed to explore ethnic differences in the socio-cultural barriers to ACP discussion, and their influence on the optimal timing of initiating ACP discussions. More importantly, studies are needed to understand how the impact of patient, caregiver, socio-cultural, and timing barriers can be mitigated, in order for ACP discussions to move beyond the contemplation stage to the necessary discussion phase among both patients and their caregivers.

## Figures and Tables

**Table 1 ijerph-18-07150-t001:** Characteristics of the patient–caregiver dyads who underwent ACP counseling (*N* = 20).

	Mean ± SD	*N* (%)
**Patient**		
Age (years)	75.9 ± 7.5	
Race:		
Chinese		17 (85)
Indian		3 (15)
Gender:		
Female		8 (40)
Male		12 (60)
Education (years)	10.1 ± 4.2	
Charlson Comorbidity Index, age-adjusted	4.6 ± 1.0	
Barthel Index (range 0–100)	98.2 ± 4.6	
Lawton and Brody’s IADL (range 0–23)	14.9 ± 3.6	
CMMSE (range 0–28)	22.8 ± 2.7	
Zarit Burden Interview (range 0–88)	22.8 ± 12.1	
Dementia Etiology:		
Alzheimer’s Dementia		15 (75)
Vascular Dementia		1 (5)
Mixed Alzheimer’s and Vascular Dementia		1 (5)
Alzheimer’s Dementia with Stroke Disease		1 (5)
Frontotemporal Dementia		1 (5)
Post Traumatic		1 (5)
**Caregiver**		
Age (years)	57.3 ± 17.3	
Gender:		
Female		7 (35)
Male		13 (65)
Education (years)	13.9 ± 4.1	
Relationship with Patient:		
Spousal Caregiver		8 (40)
Adult Child		11 (55)
Other		1 (5)
Co-residency:		
Yes		17 (85)
No		3 (15)

IADL = Instrumental Activities of Daily Living; CMMSE = modified Chinese Mini-Mental State Examination.

**Table 2 ijerph-18-07150-t002:** Patient–caregiver dyads’ perceptions of ACP.

Perceptions of ACP	Pre-ACP Counseling (*N* = 20)			Post-ACP Counseling (*N* = 19)		
	Patient *n* (%)	Caregiver *n* (%)	*p* Value	Patient *n* (%)	Caregiver *n* (%)	*p* Value
Important to talk about illness	17 (85)	20 (100)	0.231	15 (78.9)	19 (100)	0.105
Would like further discussion	11 (55)	17 (85)	0.082	13 (68.4)	15 (78.9)	0.714
Acknowledge barriers to communication	1 (5)	9 (45)	0.008	2 (10.5)	11 (57.9)	0.005
Useful in understanding complications of illness				12 (63.2)	18 (94.7)	0.042
Helpful in exploring preferences for future treatment				13 (68.4)	15 (78.9)	0.714

**Table 3 ijerph-18-07150-t003:** Barriers to advance care planning (ACP).

Themes	Factors
Patient-related factors	Transference of decision making to others Poor cognition and perceived lack of understanding Lack of inclination to plan for the future
Caregiver-related factors	Perceived negative impact on patient Caregiver’s discomfort Confidence in congruent decision-making
Socio-cultural factors	Perceived ACP discussion as a taboo topic Superstitious beliefs surrounding ACP discussions Spirituality and religious beliefs
Inappropriate timing of discussion	Good physical health state

## Data Availability

The data presented in this study are available on request from the corresponding author.

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
