# Peer review of "Exploring Differential Perceptions and Barriers to Advance Care Planning in Dementia among Asian Patient–Caregiver Dyads—A Mixed-Methods Study"

_ijerph, 2021, doi:10.3390/ijerph18137150_

Round 1

Reviewer 1 Report

Thank you for the opportunity to review the manuscript, "Barriers to communications for the twin taboos of dementia and advance care planning: An Asian memory clinic perspective."

This manuscript is very interesting and at a very good level. This study is very well written and delivered and the outcomes are clearly explained and explored in the discussion. I have read this manuscript carefully and found no severe methodological problems and significant mistakes. Well done.

I recommend minor comments.

I suggest considering putting Asian patient-caregiver dyads in the title instead of An Asian memory clinic.

Several of the articles older than 5 years cited in the background/discussion. It would be helpful to include more recent citations, especially as ACP has garnered much attention in research recently.

Author Response

We thank Reviewer 1 for the insightful comments and have provided the replies below. 

Reviewer 1 Comments:

Thank you for the opportunity to review the manuscript, "Barriers to communications for the twin taboos of dementia and advance care planning: An Asian memory clinic perspective."

This manuscript is very interesting and at a very good level. This study is very well written and delivered and the outcomes are clearly explained and explored in the discussion. I have read this manuscript carefully and found no severe methodological problems and significant mistakes. Well done.

I recommend minor comments.

Comment 1:

I suggest considering putting Asian patient-caregiver dyads in the title instead of An Asian memory clinic.

Response 1:

We appreciate the insightful suggestion and have included Asian patient-caregiver dyads in our Title, which we have also amended to better reflect our study.

Comment 2:

Several of the articles older than 5 years cited in the background/discussion. It would be helpful to include more recent citations, especially as ACP has garnered much attention in research recently.

Response 2:

We appreciate the reviewer’s comments and have revised the manuscript in the Introduction and Discussion sections to include recent citations on advance care planning in dementia. We have also updated the References section accordingly to reflect the changes in citations. We decided to retain citations of the older seminal papers vis-à-vis the more recent publications, so as to allow deeper appreciation of the salience of the study findings even at the present time. For instance, certain elements of patients’ perspectives of ACP have remained consistent over the years, such as ACP helping to relieve their loved ones of the burden of decision-making and allowing them to exercise control over their health (Page 1, Paragraph 1). Conversely, similar barriers to ACP in dementia were highlighted by both earlier and recent research publications (Page 2, Paragraph 2).

Reviewer 2 Report

It is a well-organized exploratory work, with expected results.

I believe that it could be enriched with the use of questionnaires that would allow a better knowledge of cognitive levels (e.g. MoCA) and the use of other instruments that assess decision-making ability.

Author Response

We thank Reviewer 2 for the insightful comments and provide the replies below.

Reviewer 2 Comments:

It is a well-organized exploratory work, with expected results.

I believe that it could be enriched with the use of questionnaires that would allow a better knowledge of cognitive levels (e.g., MoCA) and the use of other instruments that assess decision-making ability.

Response:

We thank the reviewer for the insightful comments. The study participants were recruited from our Memory Clinic, where evaluation of their cognitive complaints include the administration of modified Chinese Mini-Mental State Examination (cMMSE), a locally validated questionnaire to assess their cognitive function. In our study, the mean cMMSE scores (range 0 – 28) were 22.8 + SD 2.7 which reflect their mild stage of dementia. Prior to recruitment, primary physician would assess participant’s abilities to participate in ACP discussion and recruit them based on their clinical judgment.

Reviewer 3 Report

Dear authors, congratulate you on the topic and approach to the research.

However, it has essential gaps in its writing, which do not fully evaluate the investigation's internal coherence. In addition, it requires significant changes to the references.

I am attaching the comments.

Summary: it does not clearly define the objective of the study. Nor does it determine the type of paradigm, type of study, study population, sample, inclusion, or exclusion criteria. Type of data analysis or respect for ethical considerations.

Keywords: must be capitalized as in MeSH and separated by semicolons.

Introduction: state of the art must update. In addition, the referential framework needs the main actors such as WHO. In this way, define the object of study and justify its development.

Material and method: no paradigm, type of study, or design is declared. In population, we do not speak of the total population or sample design. The construction or validation of a quantitative or qualitative instrument specifies the qualitative interview only names in figure 1. Furthermore, specify how ethical considerations are respected.

Results: the titles of the tables need the N and year. It delivers a large number of products that do not specify in the methodology. The methodology must establish the order of the variables and categories studied, thus the following order in presenting results and subsequent discussion.

Discussion: larger size is expected compared to the amount of data collected. It recommends using subtitles and organize according to results to improve understanding.

Conclusion: they must base on findings and objectives set to achieve.

References: only 28.6% of the references are from the last five years, even 40% with more than ten years.

It should improve throughout the entire article, first updated, and second, leaving only those that contribute to the research, eliminating many references only used once and accompanied by another reference.

Author Response

We thank Reviewer 3 for the insightful comments and provide the replies below.

Reviewer 3 Comments:

Dear authors, congratulate you on the topic and approach to the research.

However, it has essential gaps in its writing, which do not fully evaluate the investigation's internal coherence. In addition, it requires significant changes to the references.

I am attaching the comments.

Comment 1:

Summary: it does not clearly define the objective of the study. Nor does it determine the type of paradigm, type of study, study population, sample, inclusion, or exclusion criteria. Type of data analysis or respect for ethical considerations.

Response 1:

We appreciate the reviewer’s comments and suggestions. We have made amendments to the manuscript accordingly to improve it and detailed the item-by-item responses below. 

Comment 2:

Keywords: must be capitalized as in MeSH and separated by semicolons.

Response 2:

We have made corrections to Keywords as suggested and make them inclusive to match the title and methodology.

Comment 3:

Introduction: state of the art must update. In addition, the referential framework needs the main actors such as WHO. In this way, define the object of study and justify its development.

Response 3:

We have amended the Introduction to include the following:

- an up-to-date consensus definition of ACP from a Multidisciplinary Delphi Panel of international experts

- we explained how collectivist Asian culture may pose barriers to ACP, supported by recent scientific publications

- we included a paragraph on how ACP was introduced to Singapore 10 years ago and the development of the national ACP program subsequently over the years, to form the basis of why we conduct the ACP study in persons with dementia

Comment 4:

Material and method: no paradigm, type of study, or design is declared. In population, we do not speak of the total population or sample design. The construction or validation of a quantitative or qualitative instrument specifies the qualitative interview only names in figure 1. Furthermore, specify how ethical considerations are respected.

Response 4:

We have included a subtitle, Study Design, under Materials and Methods to explain our study design on Page 3, Paragraph 2: A parallel mixed-method study design was conducted, whereby the quantitative and qualitative data were collected concurrently. Mixed methods research, in its comprehensive approach and employment of diverse data sources, yield a broader picture and more comprehensive support for validity in the investigation of underlying paradigm such as ACP in “real-world” practice. In our study, we wanted to obtain different yet complementary data to examine the perceptions of PWD and their respective caregivers on ACP, which was then a novelty in Singapore. The central phenomenon to be explored was the perceived barriers to the initiation of ACP discussions in PWD and their caregivers in a memory clinic setting in Singapore”.   

With regards to ethical considerations, patients were first assessed by primary physicians who assessed clinically if patients were able to participate in ACP discussions. Study team would then approach patient-caregiver dyads to recruit them for the study. We have written on Page 3, Paragraph 4 the following: “Consent was obtained from caregivers or legally acceptable representative and verbal assent was obtained from the patients. Conduct of the study was approved by the institutional review board of the National Healthcare Group (NHG DSRB Ref: 2010/00396)”.  

Comment 5:

Results: the titles of the tables need the N and year. It delivers a large number of products that do not specify in the methodology. The methodology must establish the order of the variables and categories studied, thus the following order in presenting results and subsequent discussion.

Response 5:

We have included the N in the titles of Table 1 and 2. We have also amended Table 1 to depict the characteristics of our patient-caregiver dyads clearly. We have revised the aims and explained the study design to achieve a coherent flow leading to the results of the study.

Comment 6:

Discussion: larger size is expected compared to the amount of data collected. It recommends using subtitles and organize according to results to improve understanding.

Response 6:

We appreciate the reviewer’s suggestions and have organized the Discussion section by the results of our study.

Comment 7:

Conclusion: they must base on findings and objectives set to achieve.

Response 7:

We appreciate the reviewer’s excellent suggestion. We have made revisions to the conclusion to achieve coherence with the overall study objectives and findings.

Comment 8:

References: only 28.6% of the references are from the last five years, even 40% with more than ten years. It should improve throughout the entire article, first updated, and second, leaving only those that contribute to the research, eliminating many references only used once and accompanied by another reference.

Response 8:

We appreciate the reviewer’s comments and have revised the manuscript in the Introduction and Discussion sections to include recent citations on ACP and ACP in dementia, and eliminate references used once that may not contribute significantly to the research. We have also updated the References section accordingly to reflect the changes in citations.

We decided to retain citations of the older seminal papers vis-à-vis the more recent publications, so as to allow deeper appreciation of the salience of the study findings even at the present time. For instance, certain elements of patients’ perspectives of ACP have remained consistent over the years, such as ACP helping to relieve their loved ones of the burden of decision-making and allowing them to exercise control over their health (Page 1, Paragraph 1). Conversely, similar barriers to ACP in dementia were highlighted by both earlier and recent research publications (Page 2, Paragraph 2)

Round 2

Reviewer 3 Report

Dear authors, I hope you are enjoying good health with your family. Always, it is excellent news the authors send the improved article. However, this needs some changes yet.

Summary: does not define the type of paradigm, study population, sample, inclusion or exclusion criteria, or respect for ethical considerations.

Introduction: although state of the art has been updated, the reference framework still needs main actors such as the WHO.

Material and method: no paradigm declared.

In population, we continue without considering the total population or the sample design. Furthermore, it remains unspecified how ethical considerations are respected, informed consent is only part of this, and it is not enough.

References: now, 58.6% of the references are from the last five years, and only 16% with more than ten years.

The number of references that appear only once in the text continues to call my attention, thus completing 70 (41 does not appear). The inclusion of main actors such as WHO is missing. Remember that this is not the vision of a country or continent but rather an international one. 

Author Response

We thank the reviewer for the insightful comments and suggestions. Kindly find item-by-item response below. 

Comment 1:

Summary: does not define the type of paradigm, study population, sample, inclusion or exclusion criteria, or respect for ethical considerations.

Response 1:

We thank the reviewer for the insightful comments and suggestions. We have indicated the research paradigm in the Materials and Methods section under Study Design (with more explanation under Comment 3 below) and included the study design of parallel mixed-methods study in the abstract. The study population was patients with mild dementia and their respective caregivers. We included English-speaking patient-caregiver dyads and excluded patients at moderate or severe stage of dementia and those with significant mood or behavioral issues. We have included exclusion criteria in the abstract as suggested. We have explained in greater detail the ethical considerations for the study under Study Design on Page 3 Paragraph 5 given the limited word count of 200 words (which we have attained) for the abstract.

Comment 2:

Introduction: although state of the art has been updated, the reference framework still needs main actors such as the WHO.

Response 2:

We appreciate the reviewer’s valuable suggestion and have amended the introduction to include World Health Organisation’s Global Dementia Observatory (GDO) Framework on Page 1, Paragraph 1: ACP was included in the World Health Organisation’s (WHO) Global Dementia Observatory (GDO) Framework, which strives to increase awareness in dementia as a public health priority and to advocate for action at national and international levels. One of the global action areas under this framework was to improve end-of-life care in persons with dementia (PWD) by promoting awareness on ACP, respecting the values and preferences of PWD, and empowering PWD to make choices about their care.

Comment 3:

(a) Material and method: no paradigm declared. In population, we continue without considering the total population or the sample design.

(b) Furthermore, it remains unspecified how ethical considerations are respected, informed consent is only part of this, and it is not enough.

Response 3:

(a) We thank the reviewer for the comment. We adopted the mixed methods approach associated with pragmatic paradigm as described by Creswell (2003),1 where our approach to conducting this research in ACP is that of mixing data collection methods and data analysis procedures within the research process. Our research paradigm is mixed methods research which is considered the third methodological or research paradigm (after quantitative and qualitative), and is a research design with philosophical assumptions and methods of inquiry.2 The premise for our research paradigm is that the use of both quantitative and qualitative approaches in combination will provide a better understanding of the central phenomenon of ACP rather than either approach alone. We have indicated our research paradigm of mixed methods research on Page 3, Paragraph 3 under Study Design: Mixed methods research is underpinned in the pragmatic research paradigm; its comprehensive approach and employment of diverse data sources yield a broader picture and more comprehensive support for validity in the investigation of an underlying phenomenon such as ACP in “real-world” practice.

1Creswell, J.W. Research Design: Qualitative, Quantitative, and Mixed Method Approaches. 2nd Edition, Sage Publications, Inc., Thousand Oaks, 2003.

2Gunasekare, D.U. Mixed research method as the third research paradigm: a Literature review. International Journal of Science and Research 2013.

(b) We appreciate the reviewer’s comment and made further amendments to our manuscript on Page 3 Paragraph 5 to explain the various ethical considerations on top of informed consent: Participant information sheets were given to eligible patient-caregiver dyads, as assessed by their primary physicians, prior to consent-taking and participation in the study was entirely voluntary. Informed consent was obtained from patient-caregiver dyads in the comfort of a private room within the hospital. Both patients and caregivers were encouraged to ask questions and clarify any doubts that they might have prior to consenting to the study. All study participants were assigned unique code numbers to maintain anonymity and confidentiality. All survey data and interviews were de-identified and stored securely with restricted access by study group. Conduct of the study was approved by the institutional review board of the National Healthcare Group (NHG DSRB Ref: 2010/00396).

Comment 4:

References: now, 58.6% of the references are from the last five years, and only 16% with more than ten years. The number of references that appear only once in the text continues to call my attention, thus completing 70 (41 does not appear). The inclusion of main actors such as WHO is missing. Remember that this is not the vision of a country or continent but rather an international one. 

Response 4:

We thank the reviewer for the comments and we have included WHO Global Dementia Observatory Framework, as written in Response 1 above, to emphasise WHO’s call for international action on provision of good quality dementia care with ACP being one of the action plans.